# Genome-wide mapping using new AFLP markers to explore intraspecific variation among pathogenic *Sporothrix* species

**Jamile Ambrósio de Carvalho**[1,2], **Ferry Hagen**[3,4,5], **Matthew C. Fisher**[6], **Zoilo Pires de Camargo**[1,2], **Anderson Messias Rodrigues**[2]*

**1** Departament of Medicine, Discipline of infectious Diseases, Federal University of São Paulo (UNIFESP), São Paulo, Brazil, **2** Laboratory of Emerging Fungal Pathogens, Department of Microbiology, Immunology, and Parasitology, Discipline of Cellular Biology, Federal University of São Paulo (UNIFESP), São Paulo, Brazil, **3** Westerdijk Fungal Biodiversity Institute, Utrecht, The Netherlands, **4** Department of Medical Microbiology, UMC Utrecht, Utrecht, The Netherlands, **5** Laboratory of Medical Mycology, Jining No. 1 People's Hospital, Jining, Shandong, Peoples Republic of China, **6** MRC Centre for Global Infectious Disease Analysis, School of Public Health, Imperial College London, London, United Kingdom

* amrodrigues.amr@gmail.com

**Data Availability Statement:** All relevant data are within the manuscript and its Supporting Information files.

## Abstract

Sporotrichosis is a chronic subcutaneous mycosis caused by *Sporothrix* species, of which the main aetiological agents are *S. brasiliensis*, *S. schenckii*, and *S. globosa*. Infection occurs after a traumatic inoculation of *Sporothrix* propagules in mammals' skin and can follow either a classic route through traumatic inoculation by plant debris (e.g., *S. schenckii* and *S. globosa*) or an alternative route through zoonotic transmission from animals (e.g., *S. brasiliensis*). Epizootics followed by a zoonotic route occur in Brazil, with Rio de Janeiro as the epicenter of a recent cat-transmitted epidemic. DNA-based markers are needed to explore the epidemiology of these *Sporothrix* expansions using molecular methods. This paper reports the use of amplified-fragment-length polymorphisms (AFLP) to assess the degree of intraspecific variability among *Sporothrix* species. We used whole-genome sequences from *Sporothrix* species to generate 2,304 virtual AFLP fingerprints. *In silico* screening highlighted 6 primer pair combinations to be tested *in vitro*. The protocol was used to genotype 27 medically relevant *Sporothrix*. Based on the overall scored AFLP markers (97–137 fragments), the values of polymorphism information content ($PIC$ = 0.2552–0.3113), marker index ($MI$ = 0.002–0.0039), effective multiplex ratio ($E$ = 17.8519–35.2222), resolving power ($Rp$ = 33.6296–63.1852), discriminating power ($D$ = 0.9291–0.9662), expected heterozygosity ($H$ = 0.3003–0.3857), and mean heterozygosity ($H_{avp}$ = 0.0001) demonstrated the utility of these primer combinations for discriminating *Sporothrix*. AFLP markers revealed cryptic diversity in species previously thought to be the most prevalent clonal type, such as *S. brasiliensis*, responsible for cat-transmitted sporotrichosis, and *S. globosa* responsible for large sapronosis outbreaks in Asia. Three combinations (#3 EcoRI-FAM-GA/MseI-TT, #5 EcoRI-FAM-GA/MseI-AG, and #6 EcoRI-FAM-TA/MseI-AA) provide the best diversity indices and lowest error rates. These methods make it easier to track routes of disease transmission during epizooties and zoonosis, and our DNA fingerprint assay can be further transferred between laboratories to give insights into the ecology and

**Funding:** AMR was supported by grants from the São Paulo Research Foundation (FAPESP 2017/27265-5) http://www.fapesp.br/, the National Council for Scientific and Technological Development (CNPq 433276/2018-5) http://www.cnpq.br/, and Coordination of Improvement of Higher Education Personnel (CAPES 88887.159096/2017-00) https://www.capes.gov.br/. ZPC was supported by grants from FAPESP (2018/21460-3). MCF is a CIFAR fellow in the 'Fungal Kingdoms' programme (https://www.cifar.ca/research/program/fungal-kingdom) and the UK Medical Research Council MR/R015600/1 (https://mrc.ukri.org/). The funders had no role in the study design, data collection and analysis, decision to publish, or preparation of the manuscript.

**Competing interests:** The authors have declared that no competing interests exist.

evolution of pathogenic *Sporothrix* species and to inform management and mitigation strategies to tackle the advance of sporotrichosis.

## Author summary

Sporotrichosis is a subacute or chronic infection characterized by nodular lesions of the (sub)cutaneous tissues and adjacent lymphatics. *Sporothrix brasiliensis*, *S. schenckii*, and *S. globosa* are the main agents of sporotrichosis in humans and other mammals. *Sporothrix* propagules gain entrance by traumatic implantation in the skin following two main routes of infection, which include animal transmission (e.g., cat-cat and cat-human) and plant origin. In recent years there has been a significant increase in the number of atypical and more severe cases of sporotrichosis, along with the expansion of the area of occurrence of *Sporothrix* species, such as the highly virulent *S. brasiliensis*. We investigated the usefulness of the AFLP technology, a DNA fingerprinting technique, which is based on the selective amplification of genomic restriction fragments by PCR to explore genetic diversity and population structure in *Sporothrix* during ongoing outbreaks. We report six highly effective sets of AFLP markers to discriminate *Sporothrix* at species and strain level, thus allowing tracking the spread of sporotrichosis. Adding molecular data in an outbreak response context can reveal better ways to improve public policies to contain the advance of sporotrichosis, by early detection, response, intervention, and follow-up.

## Introduction

*Sporothrix* (Ascomycota: Ophiostomatales) comprises 53 species reported in the literature [1]. Within a genus showing an essentially environmental core associated with plant debris, decaying wood, insects, and soil, only a few members have emerged in recent years with the ability to infect warm-blooded hosts [1]. *Sporothrix brasiliensis*, *S. schenckii*, and *S. globosa* are the main etiological agents of sporotrichosis in humans and other mammals. To a lesser extent, the disease is also caused by members of the *S. pallida* complex, such as *S. chilensis*, *S. mexicana*, and *S. pallida s. str.* [2, 3], or members of the *S. stenoceras* complex [4] which are usually non-virulent to mammals.

Sporotrichosis is a subacute or chronic fungal infection characterized by nodular lesions of the cutaneous or subcutaneous tissues and adjacent lymphatics, which suppurate, ulcerate and drain [5, 6]. *Sporothrix* propagules gain entrance by traumatic implantation in the skin following two main routes of infection, which include animal transmission (e.g., cat-cat and cat-human) and plant origin (i.e., classic sapronosis). In humans, cutaneous lesions develop at the site of inoculation, and dissemination can occur through the lymphatics during the first 2–3 weeks of infection [7]. Cats are highly susceptible to *Sporothrix*, and the most common clinical manifestations include multiple skin nodules and ulcers, often associated with nasal mucosa lesions and respiratory signs [8–11], which can lead to the development of severe forms that are difficult to treat and may lead to the death of animals [12, 13]

Infections transmitted via either animal or plant vectors usually escalate to outbreaks or epidemics [14]. Transmission routes of *Sporothrix* vary among species and host populations. Understanding how transmission spreads following host shifts is of major importance when considering the emergence of *Sporothrix* in humans and animals. For example, *S. schenckii* and *S. globosa* are cosmopolitan pathogens that appear to be widespread environmentally

following traumatic inoculation of contaminated plant material [14]. This route affects specific occupational populations, including agricultural workers, florists and gardeners, and was termed "Gardner's disease" [15] or reed toxin [16]. However, in the alternative route the highly virulent offshoot *S. brasiliensis* has spread successfully via animal horizontal transmission or zoonotic transmission in South America, so it seems that the sapronotic route plays a minor role during outbreaks caused by *S. brasiliensis*. Shifts from plant to animal transmission have led to the emergence of sporotrichosis in Brazil, and host jumping is an important feature among the Ophiostomatales, distinguishing cat-transmitted sporotrichosis as an occupation-independent disease [14].

The disease has been reported around the world, mainly in areas with high moisture and temperature [17]. The global burden of sporotrichosis is >40,000 cases [18], and the regions where the disease is endemic include Latin America (especially Brazil, Colombia, Mexico, Peru, and Venezuela) [19], Asia (especially China, India, and Japan) [20] and Africa [21]. In Brazil, recent epidemics are peculiar, especially in the South and Southeast regions, with the potential for zoonotic transmission of *S. brasiliensis*, nearly always related to cats as the main source of fungal infection for humans, dogs and other cats [22].

In Brazil, cases of sporotrichosis in humans and animals have increased significantly in recent years. The emergence of *S. brasiliensis* is associated with the appearance of atypical and more severe clinical manifestations in humans [23, 24]. This phenomenon is recent, since for decades feline sporotrichosis in Brazil occurred only as sporadic, self-limiting clusters [25, 26]. However, the current outbreak of feline sporotrichosis due to *S. brasiliensis* in South and Southeast Brazil has risen to epidemic status, creating a public health emergency of international concern because of the potential of zoonotic transmission [14, 22, 27]. Remarkably, *S. brasiliensis* has been showing spatial expansion, with a wave of expansion tracking northwards through Northeast Brazil over the last five years [1].

Tracking the movement of *Sporothrix* during outbreaks to understand the epidemiological processes driving population expansion is needed to understand epidemic sporotrichosis. However, researchers lack powerful genome-wide markers with which to address the epidemiology of *Sporothrix*. The use of amplified fragment length polymorphisms (AFLP) is one of the most informative and cost-effective DNA fingerprinting technologies applicable for any organism, without the need for prior sequence knowledge [28]. This technique provides an effective means of genotyping using a highly discriminatory panel of genome-wide markers, which is helpful in many areas of population genetics [29]. In fungi, AFLP markers have been successfully applied in studies of *Cryptococcus* spp., *Candida* spp., *Histoplasma capsulatum*, *Aspergillus fumigatus*, *Fusarium oxysporum*, *Phytophthora pinifolia*, *Monilinia fructicola*, *Fonsecaea* spp. [30–38] and *Sporothrix* spp., being considered appropriate for investigating genetic variation [20, 39, 40]. However, in some cases, failure to recover the correct tree topology displaying paraphyletic groups [20] associated with the description of no correlation between AFLP genotypes and the geographical origins of isolates [39] or clinical pictures [39, 40] led us to improve the design of these genome-wide markers, aiming finer-scale epidemiological patterns.

AFLP recognizes genetic differences between any two fungal genomes using a combination of restriction enzyme digestion of genomic DNA and PCR amplification. A pair of AFLP primers is sufficient to generate complex DNA fingerprints, and different sets of primers will yield unique profiles. Selectivity is achieved by designing primers that anneal specifically to the adaptor and the recognition site and carry one or more arbitrary chosen nucleotides at the 3′ end, which reduces the number of genomic fragments being amplified and analyzed. By using AFLP, polymorphisms can be detected by (i) a mutation in the restriction site for enzymes, (ii) a mutation in the sequence corresponding to the selective bases (primer extension), and (iii) a

deletion/insertion within the amplified fragment. To this end, we took advantage of *Sporothrix* genomes available in the GenBank to optimize, through extensive *in silico* analysis, the AFLP technique by targeting selective bases that can best answer questions related to epidemiology, genetic diversity and population structure in *Sporothrix* species. Once developed, the use of AFLP patterns of medically relevant *Sporothrix* as well as environmental species was compared to determine their genetic relationship and to explore levels of intraspecific diversity, with a focus amongst species involved in outbreaks such as *S. brasiliensis*. To achieve higher resolution, different primer combinations were tested *in vitro* to determine the best combination to be used for a focal *Sporothrix* species. Here, we report new AFLP primer combinations that will be used to test epidemiological and evolutionary hypotheses in *Sporothrix* and as high-throughput technology for assessing the degree of intraspecific variability in *Sporothrix* species during ongoing outbreaks of this disease.

## Methods

### Ethics approval

All *Sporothrix* strains used in this study belong to the culture collection of the Federal University of São Paulo (UNIFESP), Paulista School of Medicine (EPM), and were described earlier [41–44]. The protocol was approved by the Ethics in Research Committee of the Federal University of São Paulo under protocol number 2443270218.

### Fungal strains

This study included 27 *Sporothrix* isolates (*S. brasiliensis*, n = 9; *S. schenckii*, n = 8; *S. globosa*, n = 6; *S. mexicana*, n = 2; *S. chilensis*, n = 1, and *S. pallida* n = 1), obtained from clinical lesions of patients with varying degrees of disease severity (n = 20), from animals (n = 3) or from environmental sources (n = 4). The isolates were identified down to species level by comparison with the descriptions presented by Rodrigues *et al.* [41–44], and were stored at room temperature in slant cultures on Sabouraud dextrose agar (Difco Laboratories, Detroit, MI, USA) [45].

### DNA extraction

Total DNA was obtained and purified directly from 14-day-old monosporic colonies on Sabouraud slants by following the Fast DNA kit protocol (MP Biomedicals, Irvine, CA, USA), as previously described [44]. All isolates were characterized down to species level using a *Sporothrix* species-specific PCR targeting the gene encoding calmodulin, as described before [46]. Reference strains representing the main phylogenetic groups in *Sporothrix* were included in all experiments (Table 1).

### Phylogenetic and haplotype analysis

Genetic relationships for the calmodulin encoding gene sequences (exons 3–5) were investigated by phylogenetic analysis using the neighbor-joining (NJ), maximum likelihood (ML), and maximum parsimony (MP) methods. Phylogenetic trees were constructed with MEGA7 [49]. Evolutionary distances were computed using the Kimura 2-parameter distance [50] (for NJ and ML analysis), and the robustness of branches was assessed by bootstrap analysis of 1,000 replicates [51].

The nucleotide ($\pi$), as well as the haplotype (*Hd*) diversities [52], were estimated using DnaSP version 6 [53]. Haplotype network analysis was carried out using the median-joining method [54], implemented in NETWORK v4.6.1.0 (Fluxus Technology, Suffolk, UK), and was

**Table 1. Strains, species, source, origin, haplotypes, and GenBank accession numbers of *Sporothrix* spp. isolates used in this study.**

| Isolate code | Other Code | Species | Source | Origin | CAL hap[a] | GenBank | Reference |
|---|---|---|---|---|---|---|---|
| Ss09 | - | *S. brasiliensis* | Human | Brazil | H1 | KC693833 | [16] |
| Ss43 | - | *S. brasiliensis* | Human | Brazil | H2 | JX077112 | [13] |
| Ss53 | CBS 132989 | *S. brasiliensis* | Feline | Brazil | H1 | KC693846 | [16] |
| Ss55 | - | *S. brasiliensis* | Human | Brazil | H1 | KC693847 | [16] |
| Ss94 | - | *S. brasiliensis* | Human | Brazil | H1 | KF943664 | [15] |
| IPEC 16919 | - | *S. brasiliensis* | Human | Brazil | H1 | AM116898 | [1–3] |
| IPEC 16490[T] | CBS 120339 | *S. brasiliensis* | Human | Brazil | H1 | AM116899 | [1–3] |
| Ss256 | CBS 133015 | *S. brasiliensis* | Feline | Brazil | H1 | KC693889 | [16] |
| Ss261 | - | *S. brasiliensis* | Human | Brazil | H1 | KC693894 | [16] |
| Ss01 | CBS 132961 | *S. schenckii* | Feline | Brazil | H6 | KC693828 | [16] |
| Ss03 | CBS 132963 | *S. schenckii* | Human | Brazil | H6 | JX077117 | [13] |
| Ss04 | - | *S. schenckii* | Human | Brazil | H6 | JX077118 | [13] |
| Ss36 | - | *S. schenckii* | Human | Brazil | H7 | KC693843 | [16] |
| Ss58 | - | *S. schenckii* | Human | Brazil | H8 | KF943646 | [15] |
| Ss61 | - | *S. schenckii* | Soil | Brazil | H9 | KF561244 | [24] |
| Ss137 | - | *S. schenckii* | Human | Brazil | H7 | KF574462 | [24] |
| Ss143 | - | *S. schenckii* | Human | Brazil | H10 | JQ041906 | [13] |
| Ss06 | CBS 132922 | *S. globosa* | Human | Brazil | H3 | JF811336 | [13] |
| Ss41 | CBS 132923 | *S. globosa* | Human | Brazil | H4 | JF811337 | [13] |
| Ss49 | CBS 132924 | *S. globosa* | Human | Brazil | H4 | JF811338 | [13] |
| FMR 8600[T] | CBS 120340 | *S. globosa* | Human | Spain | H5 | AM116908 | [47, 48] |
| FMR 8595 | CBS 130104 | *S. globosa* | Human | Spain | H4 | AM116905 | [47, 48] |
| Ss236 | CBS 132925 | *S. globosa* | Human | Brazil | H4 | KC693877 | [47, 48] |
| FMR 9107 | CBS 120342 | *S. mexicana* | Vegetal | Mexico | H11 | AM398392 | [47, 48] |
| FMR 9108[T] | CBS 120341 | *S. mexicana* | Soil | Mexico | H11 | AM398393 | [47, 48] |
| FMR 8803 | - | *S. pallida* | Insect | China | H12 | AM398998 | [47, 48] |
| Ss469 | CBS 139891 | *S. chilensis* | Human | Chile | H13 | KP711815 | [14] |

[a]Calmodulin haplotype; IPEC, Instituto de Pesquisa Clínica Evandro Chagas, Fiocruz, Brasil; FMR, Facultat de Medicina I Ciències de La Salut, Reus, Spain; CBS, culture collection of the Westerdijk Fungal Biodiversity Institute, Utrecht, The Netherlands; All "Ss" strains belong to the culture collection of Federal University of São Paulo (UNIFESP), Paulista School of Medicine (EPM).

used to visualize differences and diversity among *Sporothrix* species sequence data. Sites containing gaps and missing data were not considered in the analysis.

## *In silico* AFLP analyses

Whole-genome sequences of nine *Sporothrix* isolates (Table 2) were analyzed *in silico* to predict AFLP markers of corresponding lengths (50–500 bp) that might be generated *in vitro* during the standard AFLP procedure. *In silico* AFLP analyses were performed using AFLPinSilico [55] and ISIF (In Silico Fingerprinting) [56]. Briefly, *Sporothrix* genomes were retrieved from the GenBank and *in silico* digested with EcoRI and MseI restriction enzymes. Afterward, a total of 256 combinations of two selective bases (EcoRI+2 and MseI+2) were used to mine a subset of selective fragments. Finally, to properly simulate the AFLP procedure, we determined the length of all the peaks of the *in silico* AFLP profile, with the addition of the adaptor and primer lengths. The number and size of the fragments were used to construct a matrix of fragments and data were visualized using heatmaps. Hierarchical cluster analysis of *in silico* AFLP

**Table 2. Genomes of *Sporothrix* species retrieved from NCBI Genome database (https://www.ncbi.nlm.nih.gov/genome) for *in silico* analysis.**

| Strain | Species | Source | Origin | INSDC[1] (WGS) | Total length | BioProjects | Reference |
|--------|---------|--------|--------|----------------|--------------|-------------|-----------|
| 5110 | *S. brasiliensis* | Feline | Brazil | AWTV01 | 33.2 Mb | PRJNA218075 | [58] |
| ATCC 58251 | *S. schenckii* | Human | USA | AWEQ01 | 32.5 Mb | PRJNA217088 | [59] |
| 1099–18 | *S. schenckii* | Human | USA | AXCR01 | 32.5 Mb | PRJNA218070 | [58] |
| SsMS1 | *S. schenckii* | Human | Colombia | PGUU01 | 32.6 Mb | PRJNA401003 | [60] |
| SsEM7 | *S. schenckii* | Human | Colombia | NTMI01 | 32.8 Mb | PRJNA401003 | [60] |
| CBS 120340 | *S. globosa* | Human | Spain | LVYW01 | 33.4 Mb | PRJNA315855 | [61] |
| SS01 | *S. globosa* | Human | China | LVYX01 | 33.4 Mb | PRJNA315862 | [61] |
| SPA8 | *S. pallida* | Soil | Spain | JNEX02 | 37.8 Mb | PRJNA248334 | [62] |
| RCEF 264 | *S. insectorum* | Insect | China | AZHD01 | 34.7 Mb | PRJNA72727 | [63] |

[1]International Nucleotide Sequence Database Collaboration (INSDC; http://www.insdc.org/)

profiles was performed using Heatmapper [57], based on average linkage and Euclidean distance applied to each row-cluster.

## Genotyping by AFLP fingerprinting

Digestion of DNA, adapter ligation, non-selective, and selective amplifications were carried out *in vitro* as described previously by Vos *et al.* [64] with some modifications. Briefly, 200 ng of *Sporothrix* genomic DNA was digested by EcoRI and MseI (New England Biolabs, Ipswich, MA) and ligated to EcoRI and MseI adapters simultaneously (Integrated DNA Technologies, USA) [64]. A preselective amplification was performed with EcoRI+0 and MseI+0 primers [64]. Fluorescent AFLP was done with 6-carboxyfluorescein (FAM; blue) fluorescent dye-labeled EcoRI primer, with two selective bases (5′-GAC TGC GTA CCA ATT CNN-3′), and unlabeled MseI primer with two selective bases (5′-GAT GAG TCC TGA GTA ANN-3′). Six different combinations were chosen to evaluate the potential for genetic characterization of clinical *Sporothrix* isolates (combination #1 EcoRI-GA/MseI-AA; #2 EcoRI-AA/MseI-AA; #3 EcoRI-GA/MseI-TT; #4 EcoRI-AA/MseI-TT; #5 EcoRI-GA/MseI-AG; #6 EcoRI-TA/MseI-AA). All oligonucleotides were supplied by Integrated DNA Technologies (IDT, San Diego, CA, USA). PCR-amplified AFLP fragments were resolved by capillary electrophoresis with an ABI3100 Genetic Analyzer alongside a LIZ500 internal size standard (Applied Biosystems. Foster City, CA, USA) at the Human Genome and Stem Cell Research Center Core Facility (University of São Paulo, São Paulo, Brazil) under previously described conditions. At least two independent electropherograms for each combination of selective primers were imported in BioNumerics v7.6 (Applied Maths, St. Martens-Latem, Belgium) and analyzed to verify the ability to accurately reproduce results.

The selection of the amplified restriction products was automated, and only strong and high-quality fragments were considered. To minimize scoring errors, each electropherogram was carefully inspected to exclude doubtful peaks, setting a minimum threshold at 100 relative fluorescent units, and considering only peaks with sizes between 50 and 500 base pairs. The size of the AFLP fragments was determined by BioNumerics v7.6. Peak patterns were converted to the dominant presence (1) or absence (0) at probable fragment positions.

Relationships among *Sporothrix* specimens and taxa were evaluated employing distance-based methods, as recommended for dominant anonymous markers. Therefore, pairwise genetic distances were calculated using the band-based Jaccard's similarity coefficient combined with a "fuzzy logic" option in BioNumerics v7.6. Dendrograms were created according to the unweighted pair group mean arithmetic method (UPGMA). Branch resampling support

was conducted using the cophenetic correlation coefficient and the standard deviation was used to express the consistency of a given cluster, which determines the correlation between the dendrogram-derived similarities and the matrix similarities.

To assess the existence of topological congruence between any two AFLP dendrograms or between AFLP dendrograms and DNA-sequencing phylogeny, and the associated confidence level, Newick trees were used to calculate the congruence index (I*cong*), as described by de Vienne and colleagues [65], based on maximum agreement subtrees (MAST).

The minimum spanning tree (MST) model was used to investigate the evolutionary relationships among all the observed genotypes of medically relevant *Sporothrix* species. AFLP-derived MSTs were executed in BioNumerics v7.6 with AFLP data after conversion into a band matching table. When a set of distances is given between *n* entries, a minimum spanning tree is the tree that connects all entries in such a way that the summed distance of all branches of the tree is the shortest possible [66]. All figures were exported and treated using Corel Draw X8 (Corel, Ottawa, Canada).

## Reproducibility of AFLP markers

Reproducibility of AFLP fragment profiles was assessed by repeated digestion of DNA, adapter ligation, non-selective and selective amplification of all individuals genotyped per species [67]. A single error rate was calculated for all the samples analyzed. Error rates were determined as the percentage of loci that were mismatched between the replicate pairs [68].

## Genetic diversity and statistical analysis

To evaluate which of the six AFLP primer combinations were the most informative, the following polymorphism indices for dominant markers were calculated: polymorphic information content (*PIC*) [69], expected heterozygosity (*H*) [70], effective multiplex ratio (*E*) [71], arithmetic mean heterozygosity ($H_{avp}$) [71], marker index (*MI*) [71, 72], discriminating power (*D*) [73], and resolving power (*Rp*) [74].

## Dimensioning analysis

Principal component analysis (PCA) and multi-dimensional scaling (MDS) were used as alternative grouping methods, producing three-dimensional plots in which the entries were spread according to their relatedness. Dimensioning techniques were executed with AFLP data after conversion into a band matching table. Automated band matching was performed on all fingerprint entries within the comparison, considering minimum profiling of 5%, with the optimization and position tolerances for selecting bands set to 0.10%. Default settings were applied for PCA and MDS in BioNumerics v7.6, subtracting the average for characters [38]. All Figs were exported and treated using Corel Draw X8.

## Results

We selected 27 well-characterized *Sporothrix* isolates and used calmodulin to explore genetic diversity and intraspecific variability in our dataset. The aligned *CAL* sequences were 623 bp long, including 437 invariable characters, 167 variable parsimony-informative sites (26.8%), and 19 singletons. The clade of pathogenic *Sporothrix* species was well supported with high bootstrap values (100), including *S. brasiliensis* (clade I), *S. schenckii* (clade II), and *S. globosa* (clade III) (Fig 1A). Comparison of *CAL* sequences revealed significant polymorphisms among *S. schenckii* isolates ($\pi$ = 0.00908), whereas sequences from *S. brasiliensis* ($\pi$ = 0.00000) and *S. globosa* ($\pi$ = 0.00112) were much less diverse. The haplotype diversity was

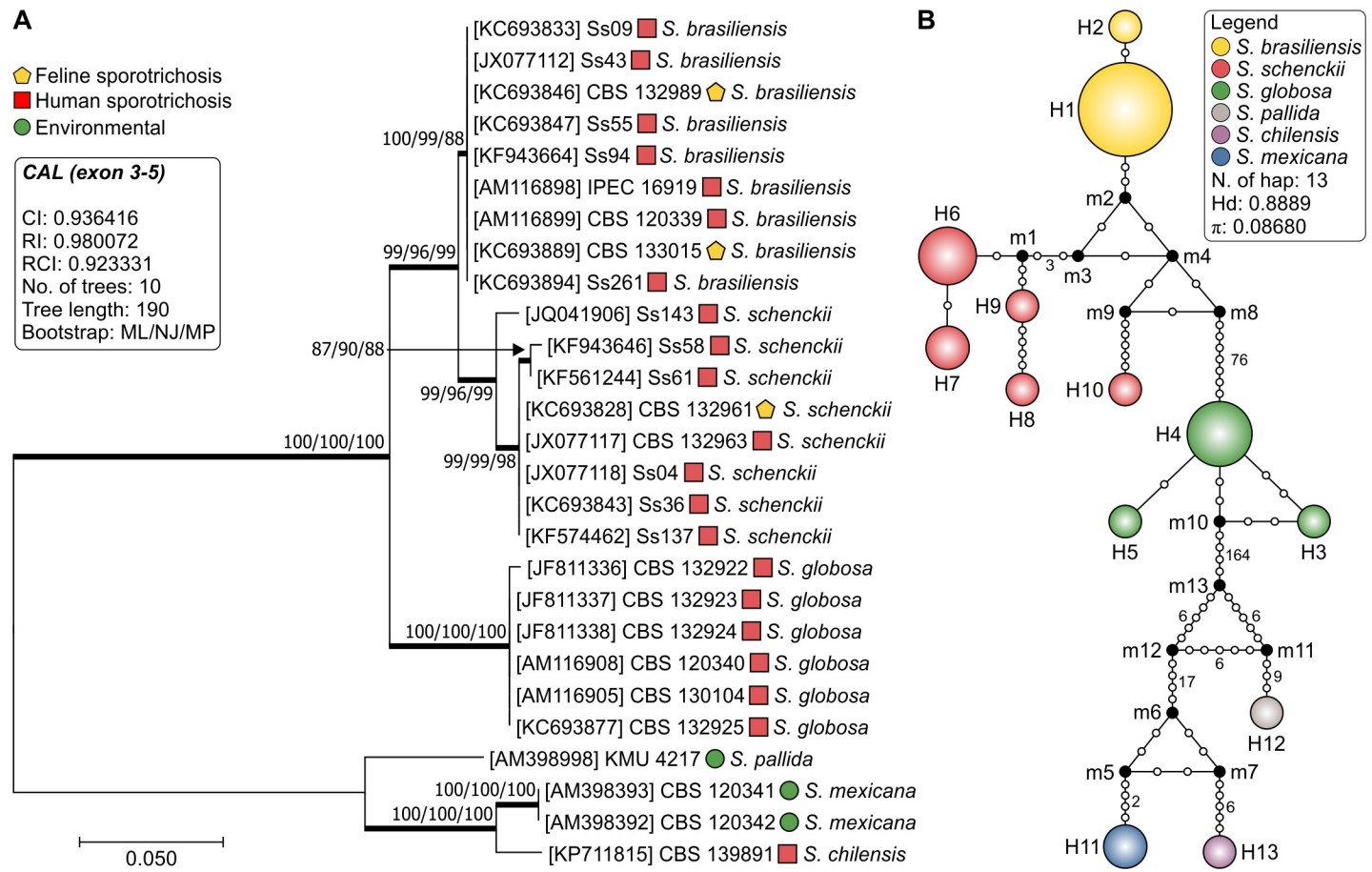

**Fig 1.** Phylogenetic tree (A), inferred using the maximum likelihood method and Kimura 2-parameter model of the calmodulin sequences of 27 strains of *Sporothrix*. Numbers close to the branches represent ML/NJ/MP respectively. Bootstraps higher than 95 based on 1000 replications are represented in bold branches. Haplotype network of *Sporothrix* (B) was done using the median-joining method. The circumference size is proportional to the frequency of haplotype. The median vectors are displayed by black dots and represent hypothetical unsampled or extinct haplotypes in the population.

assessed using the DnaSP software [53], which allowed identifying 13 distinct haplotypes. The overall values of haplotype ($Hd$ = 0.8889) and nucleotide diversities ($\pi$ = 0.08680) were high for *Sporothrix*. However, when considering only *S. brasiliensis* or *S. globosa* just two ($Hd$ = 0.2222) and three ($Hd$ = 0.6000) haplotypes were found, respectively, suggesting that *CAL* works as a suitable barcoding marker for diagnosis but may cover cryptic diversity in these species (Fig 1B). The Simpson index of diversity [75] was calculated as 0.2279.

To test the hypothesis of cryptic diversity in *S. brasiliensis* and *S. globosa*, we developed AFLP markers to explore genetic variation in these emerging agents. The first step in our strategy involved the *in silico* characterization of *Sporothrix* genomes deposited with NCBI, comprising medically relevant members as well as environmental species. AFLPinSilico and ISIF were used to scan restriction sites for EcoRI and MseI. Afterward, a subset of modified genomic fragments was created by adding EcoRI and MseI adaptors, and an enriched population of modified genetic fragments was chosen based on two selective bases for EcoRI+2 and MseI+2 primers. Therefore, 256 possible combinations were investigated for nine genomes, producing a matrix of 2,304 virtual AFLP profiles, which are presented as a heatmap in Fig 2. A remarkable diversity of fragments was generated, which ranged from 0–56 and 0–62 in 256 combinations tested in AFLPinSilico and ISIF, respectively (Fig 3, S1 Table). *Sporothrix pallida*

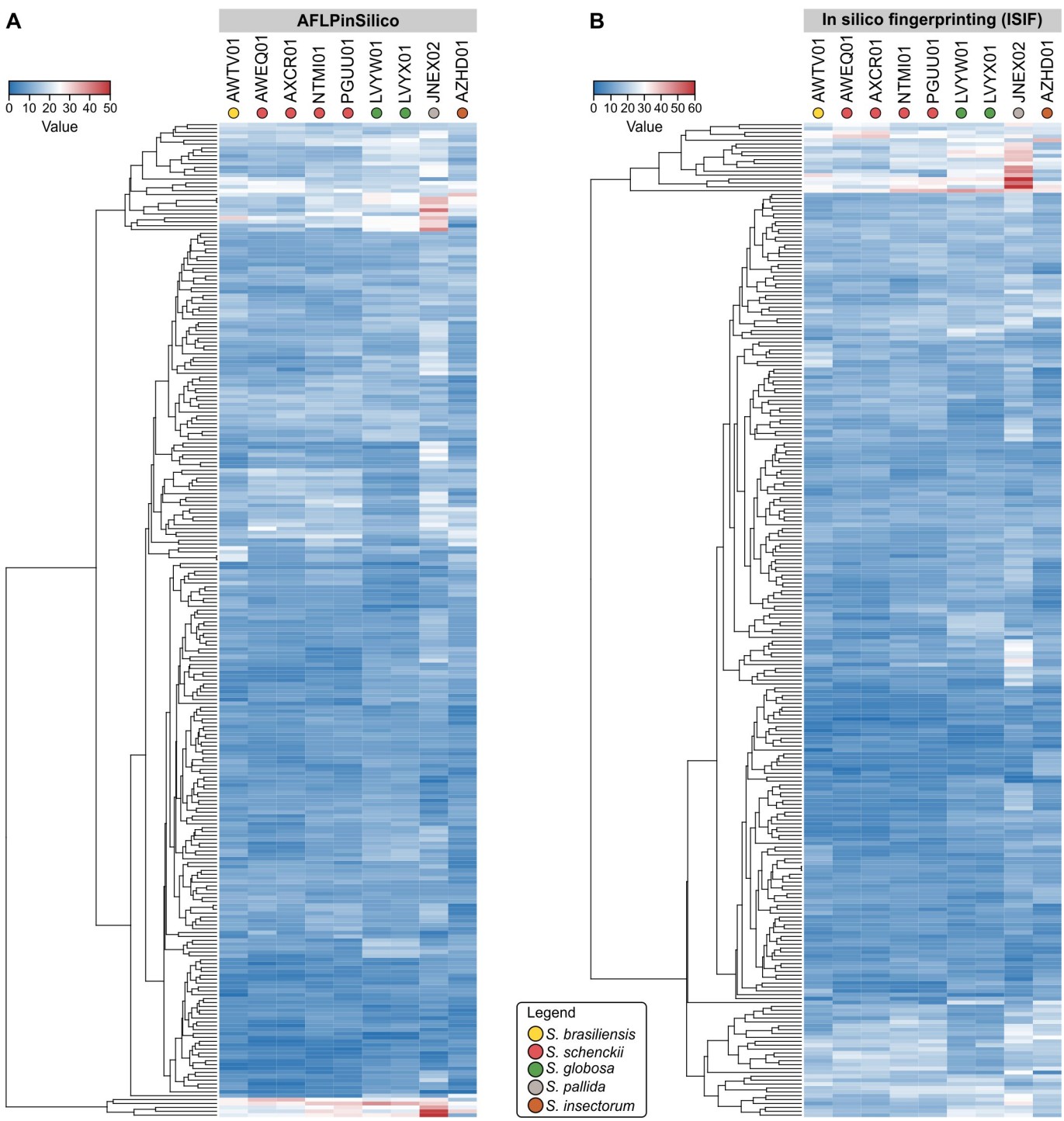

**Fig 2.** Heatmap of fragments generated by *in silico* analyses with 256 combinations of selective primer pairs in two programs–AFLPinSilico (A) and ISIF (B)–for nine genome sequences of *Sporothrix* retrieved from the GenBank. The highest numbers of fragments are represented by red shading and the lower by blue shading.

presented the largest genome core (~37.8 Mb) and it was represented in our *in silico* scan by the largest number of AFLP markers in 62.5–70.7% combinations (Pearson correlation = 0.861, $r^2$ = 0.7421, *P* = 0.00283) (Fig 3).

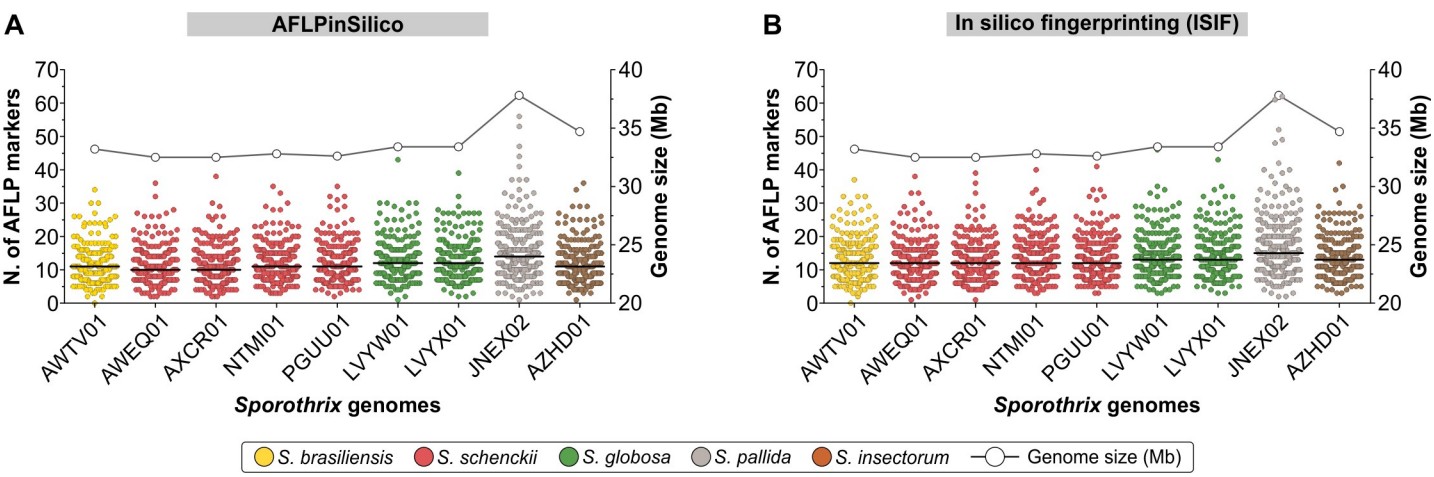

**Fig 3.** A total of 256 combinations of selective *Eco*RI+2 and *Mse*I+2 primer pairs were employed to generate 2,304 virtual AFLP profiles AFLPinSilico (A) and ISIF (B). The dots located on the left X-axis represent the number of fragments generated for each combination. The bold bar represents the average of fragments generated for all combinations. The white dots located on the right X-axis represent the genome size, estimated by whole genome sequencing.

Among all *Sporothrix* species evaluated, *S. brasiliensis* and *S. globosa* had little genetic diversity (Fig 1B), so the diversity of fragments generated for these emerging species were fundamental for the selection of putative combinations (Fig 3A). Therefore, we highlighted six combinations (#1–6) to be tested *in vitro*, which showed the highest number of polymorphic markers (i.e., number and size) with the potential to be used to speciate *Sporothrix* and explore intraspecific variation (S2 Table).

A total of 685 loci were amplified using the selective primers EcoRI+2 and MseI+2, among them 135, 137, 106, 111, 99 and 96 polymorphic fragments, for combinations #1 to #6, respectively. The averages of fragments varied per isolate for the six combinations between 19.5–32.8 for *S. brasiliensis*, 19.3–40.6 for *S. schenckii*, 15.6–31.5 for *S. globosa*, 14.5–41.5 for *S. mexicana*, 23–44 for *S. pallida* and 16–45 for *S. chilensis*. The details of marker attributes for different AFLP primer combinations are given in Table 3.

The *PIC* determined for each primer pair was both comparable between species and between markers. Overall, *PIC* values ranged from 0.2552 to 0.3145, and all markers presented high discrimination power. The highest *PIC* value was observed for primer combination 5 and the lowest was recorded for primer combination 6, indicating good diversity among the studied *Sporothrix*. Interestingly, the highest *PIC* value for *S. globosa* was obtained in combination 6 (0.3714), demonstrating the potential use of this primer pair to explore diversity in *S. globosa*. In general, *S. brasiliensis* and *S. schenckii* showed slightly higher values than *S. globosa* (Table 3).

Marker index (*MI*) as a feature of marker diversity representing the product of the effective multiplex ratio (*E*) and the arithmetic mean heterozygosity (*Havp*) was also calculated for all the primer combinations. The *MI* values ranged from 0.002 to 0.0039. The highest value (0.0039) was obtained with primer pair 5 and the lowest values (0.002) for primer pair 6. A positive correlation was observed between *MI* and *PIC* values (Pearson correlation = 0.9950171, $r^2 = 0.9901$, $P = 0.00001$). The resolving power, *Rp*, is a feature that indicates the discriminatory potential of the marker to distinguish between large numbers of genotypes. *Rp* ranged from 33.6296 to 63.1852. The highest value (63.1852) was scored with primer combination 1 and the lowest (33.6296) for primer combination 6. The *Rp* values were not positively

**Table 3. Summary of polymorphism statistics calculated for different pairs of selective primers (EcoRI+2 and MseI+2) of *Sporothrix* species.**

#1 EcoRI-GA/MseI-AA

| Species | Scored bands | *H* | *PIC* | *E* | *Havp* | *MI* | *D* | *Rp* | Error % |
|---|---|---|---|---|---|---|---|---|---|
| *S. brasiliensis* | 62 | 0.4991 | 0.3745 | 29.6667 | 0.0009 | 0.0265 | 0.7715 | 21.3333 | 1.12 |
| *S. schenckii* | 84 | 0.4995 | 0.3747 | 40.6250 | 0.0007 | 0.0302 | 0.7665 | 38.2500 | 1.23 |
| *S. globosa* | 43 | 0.3918 | 0.3151 | 31.5000 | 0.0015 | 0.0478 | 0.4641 | 9.6667 | 1.05 |
| *S. mexicana* | 45 | 0.1435 | 0.1332 | 41.5000 | 0.0016 | 0.0661 | 0.1503 | 7.0000 | 0.00 |
| Overall | 135 | 0.3857 | 0.3113 | 35.2222 | 0.0001 | 0.0037 | 0.9320 | 63.1852 | 1.04 |

#2 EcoRI-AA/MseI-AA

| Species | Scored bands | *H* | *PIC* | *E* | *Havp* | *MI* | *D* | *Rp* | Error % |
|---|---|---|---|---|---|---|---|---|---|
| *S. brasiliensis* | 56 | 0.4997 | 0.3749 | 28.6667 | 0.0010 | 0.0284 | 0.7385 | 16.8889 | 0.00 |
| *S. schenckii* | 63 | 0.4854 | 0.3676 | 26.1250 | 0.0010 | 0.0252 | 0.8285 | 26.7500 | 0.47 |
| *S. globosa* | 34 | 0.4192 | 0.3313 | 23.8333 | 0.0021 | 0.0490 | 0.5097 | 10.3333 | 0.64 |
| *S. mexicana* | 37 | 0.0267 | 0.0263 | 36.5000 | 0.0004 | 0.0132 | 0.0270 | 1.0000 | 0.00 |
| Overall | 137 | 0.3287 | 0.2747 | 28.4074 | 0.0001 | 0.0025 | 0.9570 | 52.0741 | 0.28 |

#3 EcoRI-GA/MseI-TT

| Species | Scored bands | *H* | *PIC* | *E* | *Havp* | *MI* | *D* | *Rp* | Error % |
|---|---|---|---|---|---|---|---|---|---|
| *S. brasiliensis* | 52 | 0.4974 | 0.3737 | 27.8889 | 0.0011 | 0.0296 | 0.7129 | 19.7778 | 0.00 |
| *S. schenckii* | 52 | 0.4833 | 0.3665 | 21.2500 | 0.0012 | 0.0247 | 0.8336 | 17.5000 | 0.58 |
| *S. globosa* | 28 | 0.3501 | 0.2888 | 21.6667 | 0.0021 | 0.0451 | 0.4023 | 4.0000 | 0.00 |
| *S. mexicana* | 21 | 0.0465 | 0.0454 | 20.5000 | 0.0011 | 0.0227 | 0.0476 | 1.0000 | 0.00 |
| Overall | 106 | 0.3449 | 0.2854 | 23.4815 | 0.0001 | 0.0028 | 0.9510 | 40.8889 | 0.16 |

#4 EcoRI-AA/MseI-TT

| Species | Scored bands | *H* | *PIC* | *E* | *Havp* | *MI* | *D* | *Rp* | Error % |
|---|---|---|---|---|---|---|---|---|---|
| *S. brasiliensis* | 44 | 0.4999 | 0.3750 | 21.7778 | 0.0013 | 0.0275 | 0.7557 | 16.4444 | 0.51 |
| *S. schenckii* | 52 | 0.4967 | 0.3733 | 23.8750 | 0.0012 | 0.0285 | 0.7898 | 19.7500 | 0.52 |
| *S. globosa* | 38 | 0.4038 | 0.3223 | 27.3333 | 0.0018 | 0.0484 | 0.4835 | 8.6667 | 0.00 |
| *S. mexicana* | 28 | 0.0000 | 0.0000 | 28.0000 | 0.0000 | 0.0000 | 0.0000 | 0.0000 | 0.00 |
| Overall | 111 | 0.3483 | 0.2876 | 24.9259 | 0.0001 | 0.0029 | 0.9496 | 45.6296 | 0.32 |

#5 EcoRI-GA/MseI-AG

| Species | Scored bands | *H* | *PIC* | *E* | *Havp* | *MI* | *D* | *Rp* | Error % |
|---|---|---|---|---|---|---|---|---|---|
| *S. brasiliensis* | 55 | 0.4993 | 0.3746 | 28.5556 | 0.0010 | 0.0288 | 0.7309 | 18.6667 | 0.78 |
| *S. schenckii* | 52 | 0.4958 | 0.3729 | 28.3750 | 0.0012 | 0.0338 | 0.7028 | 14.2500 | 0.00 |
| *S. globosa* | 29 | 0.4096 | 0.3257 | 20.6667 | 0.0024 | 0.0486 | 0.4933 | 10.0000 | 0.00 |
| *S. mexicana* | 28 | 0.0000 | 0.0000 | 28.0000 | 0.0000 | 0.0000 | 0.0000 | 0.0000 | 0.00 |
| Overall | 99 | 0.3908 | 0.3145 | 26.3704 | 0.0001 | 0.0039 | 0.9291 | 40.9630 | 0.29 |

#6 EcoRI-TA/MseI-AA

| Species | Scored bands | *H* | *PIC* | *E* | *Havp* | *MI* | *D* | *Rp* | Error % |
|---|---|---|---|---|---|---|---|---|---|
| *S. brasiliensis* | 49 | 0.4787 | 0.3641 | 19.4444 | 0.0011 | 0.0211 | 0.8431 | 21.7778 | 1.14 |
| *S. schenckii* | 53 | 0.4639 | 0.3563 | 19.3750 | 0.0011 | 0.0212 | 0.8669 | 22.2500 | 0.00 |
| *S. globosa* | 36 | 0.4928 | 0.3714 | 15.8333 | 0.0023 | 0.0361 | 0.8077 | 18.3333 | 0.00 |
| *S. mexicana* | 15 | 0.0644 | 0.0624 | 14.5000 | 0.0021 | 0.0311 | 0.0667 | 1.0000 | 0.00 |
| Overall | 97 | 0.3003 | 0.2552 | 17.8519 | 0.0001 | 0.0020 | 0.9662 | 33.6296 | 0.44 |

*D* = discriminating power; *E* = effective multiplex ratio; *H* = expected heterozygosity; *Havp* = mean heterozygosity; *MI* = marker index; *PIC* = polymorphism information content; *Rp* = resolving power.

correlated with *MI* (Pearson correlation = 0.4797601, r$^2$ = 0.2302, *P* = 0.335). Nevertheless, combination 6 provided the highest values of *Rp* (18.3333) and *MI* (0.0361) for *S. globosa*.

We also determined expected heterozygosity (*H*), which is defined as the probability that an individual is heterozygous for the locus in the population. It is equivalent to Nei's unbiased gene diversity ($H_S$), as adapted for dominant markers under the assumptions of Hardy-Weinberg equilibrium and the Lynch-Milligan model [76]. The overall average expected heterozygosity for *Sporothrix* species ranged between 0.3003–0.3908 (Table 4). The high combined expected heterozygosity for *S. brasiliensis* (*H* = 0.4787–0.4999) and *S. globosa* (*H* = 0.3501–0.4928) is surprising given previous reports [14, 20, 42, 44, 77–79], which showed little or no genetic variation within these pathogens based on DNA sequencing data. Indeed, the indices reported here are comparable to that of *S. schenckii* (0.4639–0.4995), a species described as more diverse than *S. brasiliensis* and *S. globosa*. Based on these criteria, the high number of loci generated for *S. brasiliensis*, *S. schenckii*, and *S. globosa* is sufficient to reveal the true fine-scale population genetic structure, and we recommend the use of combinations #5 (EcoRI--FAM-GA/MseI-AG), #6 (EcoRI-FAM-TA/MseI-AA) and #3 (EcoRI-FAM-GA/MseI-TT) to explore genetic diversity in *Sporothrix* species.

We evaluated the quality of standard fluorescent AFLP genotyping by determining the error rate and reproducibility of our datasets. The suggested and generally acceptable error rate for AFLP data ranges between 2–5% [29, 80]. AFLP electropherograms (50–500 bp) of *S. brasiliensis* (Fig 4A and 4B) and *S. globosa* (Fig 4C) show the high reproducibility of the fingerprints obtained using fluorescent capillary electrophoresis, which was crucial to reliably score AFLP markers. A fragment was considered unreliable if it showed any variability within the duplicate tested. For duplicated genotyped *Sporothrix* isolates, we observed the lowest average error rate across all markers for *S. globosa* (0.00–1.05%), *S. brasiliensis* (0.00–1.14%), and *S. schenckii* (0.00–1.23%). All the fragments scored were reproducible. Error rates never exceeded 1.23%, indicating that our protocol is highly reproducible across *Sporothrix* species, which are the main agents of human and animal sporotrichosis [80].

Typical AFLP dendrograms based on Jaccard's similarity coefficient are depicted in Fig 5. These show six well-supported clades with a similarity level ranging between 40 and 98%, with high cophenetic values (>90) in the majority of branches (S2 Table). This is in accordance with the generally applied calmodulin-based classification of Marimon *et al.* [47]. The first clade comprises *S. brasiliensis* isolates recovered from human and animal cases of sporotrichosis. It was possible to reveal cryptic diversity for *S. brasiliensis* in all datasets, with similarity levels ranging from 37.53% ± 4.06% to 51.96% ± 2.83% (S2 Table). AFLP markers could separate the *S. brasiliensis* isolates belonging to Rio Grande do Sul, which remained together in all combinations. The polymorphic fragments ranged from 38–50 fragments in the six combinations for all *S. brasiliensis* isolates, in a total of 44–62 fragments. The second clade comprised *S. schenckii*, which showed greater diversity, ranging from 27.44% ± 2.55% to 52.56% ± 1.43% (S2 Table), compared to other isolates of clinical relevance, corroborating the classic findings of DNA sequencing. Polymorphic fragments ranged from 37–77 per combination in a total of 52–84 fragments generated. The third clade comprised *S. globosa* isolates, which showed a similar pattern for the six isolates tested with cryptic diversity (42.79% ± 1.61% to 74.38% ± 1.82%), but lower than *S. brasiliensis* and *S. schenckii* isolates. The total of fragments generated for *S. globosa* varied from 28–43, with a range of 10 to 31 polymorphic fragments per combination. The remaining isolates represent members of the *S. pallida* complex, including *S. chilensis*, *S. mexicana*, and *S. pallida s. str.* In clade 4, AFLP markers varied from 14–44 fragments, and *S. pallida* isolate FMR 8803 grouped apart from the other members of the *S. pallida* complex (i.e. *S. chilensis* and *S. mexicana*) in combinations #1, #2 and #4. The two isolates of *S. mexicana* presented high genetic similarity (above 90% in 5 out of 6 combinations), generating 15–45 fragments. Polymorphic bands were absent in combinations #3 and #4, whereas in

**Table 4. Comparison among the clustering methods used for phylogenetic analysis and AFLP-derived dendrograms for combinations #1 to #6 generated by congruence index values (I*cong*).**

| Tree comparison | Leaves | MAST[a] | I*cong* | P-value | Congruent |
|---|---|---|---|---|---|
| AFLP 1 vs AFLP 2 | 27 | 13 | 1.714 | 2.19e-05 | Yes |
| AFLP 1 vs AFLP 3 | 26 | 11 | 1.479 | 0.0009 | Yes |
| AFLP 1 vs AFLP 4 | 27 | 12 | 1.583 | 0.0001 | Yes |
| AFLP 1 vs AFLP 5 | 26 | 13 | 1.748 | 1.46e-05 | Yes |
| AFLP 1 vs AFLP 6 | 26 | 13 | 1.748 | 1.46e-05 | Yes |
| AFLP 2 vs AFLP 3 | 27 | 13 | 1.714 | 2.19e-05 | Yes |
| AFLP 2 vs AFLP 4 | 27 | 14 | 1.846 | 2.79e-05 | Yes |
| AFLP 2 vs AFLP 5 | 27 | 14 | 1.846 | 2.79e-05 | Yes |
| AFLP 2 vs AFLP 6 | 27 | 13 | 1.714 | 2.19e-05 | Yes |
| AFLP 3 vs AFLP 4 | 27 | 16 | 2.110 | 4.54e-08 | Yes |
| AFLP 3 vs AFLP 5 | 26 | 12 | 1.613 | 0.0001 | Yes |
| AFLP 3 vs AFLP 6 | 26 | 11 | 1.479 | 0.0009 | Yes |
| AFLP 4 vs AFLP 5 | 27 | 15 | 1.978 | 3.56e-07 | Yes |
| AFLP 4 vs AFLP 6 | 27 | 11 | 1.451 | 0.0013 | Yes |
| AFLP 5 vs AFLP 6 | 26 | 14 | 1.882 | 1.82e-06 | Yes |
| AFLP 1 vs Cal | 26 | 11 | 1.479 | 0.0009 | Yes |
| AFLP 2 vs Cal | 27 | 12 | 1.583 | 0.0001 | Yes |
| AFLP 3 vs Cal | 26 | 10 | 1.344 | 0.0075 | Yes |
| AFLP 4 vs Cal | 27 | 23 | 3.034 | 2.46e-14 | Yes |
| AFLP 5 vs Cal | 26 | 12 | 1.613 | 0.0001 | Yes |
| AFLP 6 vs Cal | 26 | 10 | 1.344 | 0.0075 | Yes |

[a] MAST, Maximum Agreement Subtree.

combination #1 we detected seven polymorphic bands. Likewise, *S. chilensis* produced a range of 15–45 fragments, following the same clustering pattern as combinations #2 to #6.

To assess the existence of topological congruence between any two dendrograms or between dendrograms and the calmodulin phylogenetic tree, we used the congruence index (I*cong*) [65]. Multiple comparisons revealed a similar and consistent clustering pattern, as evidenced by I*cong* values and their significant associated *P*-values (Table 4). Therefore, the AFLP dendrogram and calmodulin trees are more congruent than expected by chance, supporting the use of new AFLP markers to speciate *Sporothrix* with the same confidence of DNA-sequencing methods.

The AFLP markers were used to generate pairwise genetic distance matrices based on Jaccard's similarity coefficient, which were subjected to a principal component analysis (PCA) with BioNumerics v7.6 to allow 3D graphic visualization of the relationships among each *Sporothrix* sample with those representing the same taxon. Fig 6 shows the PCA plots for combinations #1–6, and the distribution of 27 *Sporothrix* isolates among the three co-ordinates agreed with the UPGMA tree. Combinations #3, #5 and #6 revealed the highest cumulative percentage explained, with 58.7–59.2% of the variation explained by the first three components together (coordinates X, Y, and Z). The dimensioning analysis showed clearly both the high degree of intraspecific clustering and the relatively large genetic separation between any individual taxon (interspecific variation). This tight clustering supports cryptic diversity in *S. brasiliensis* and *S. globosa*. In all combinations evaluated, *S. schenckii* continued being a more diverse species than its siblings *S. brasiliensis* and *S. globosa*, in line with the higher level of intraspecific variability demonstrated in calmodulin sequencing.

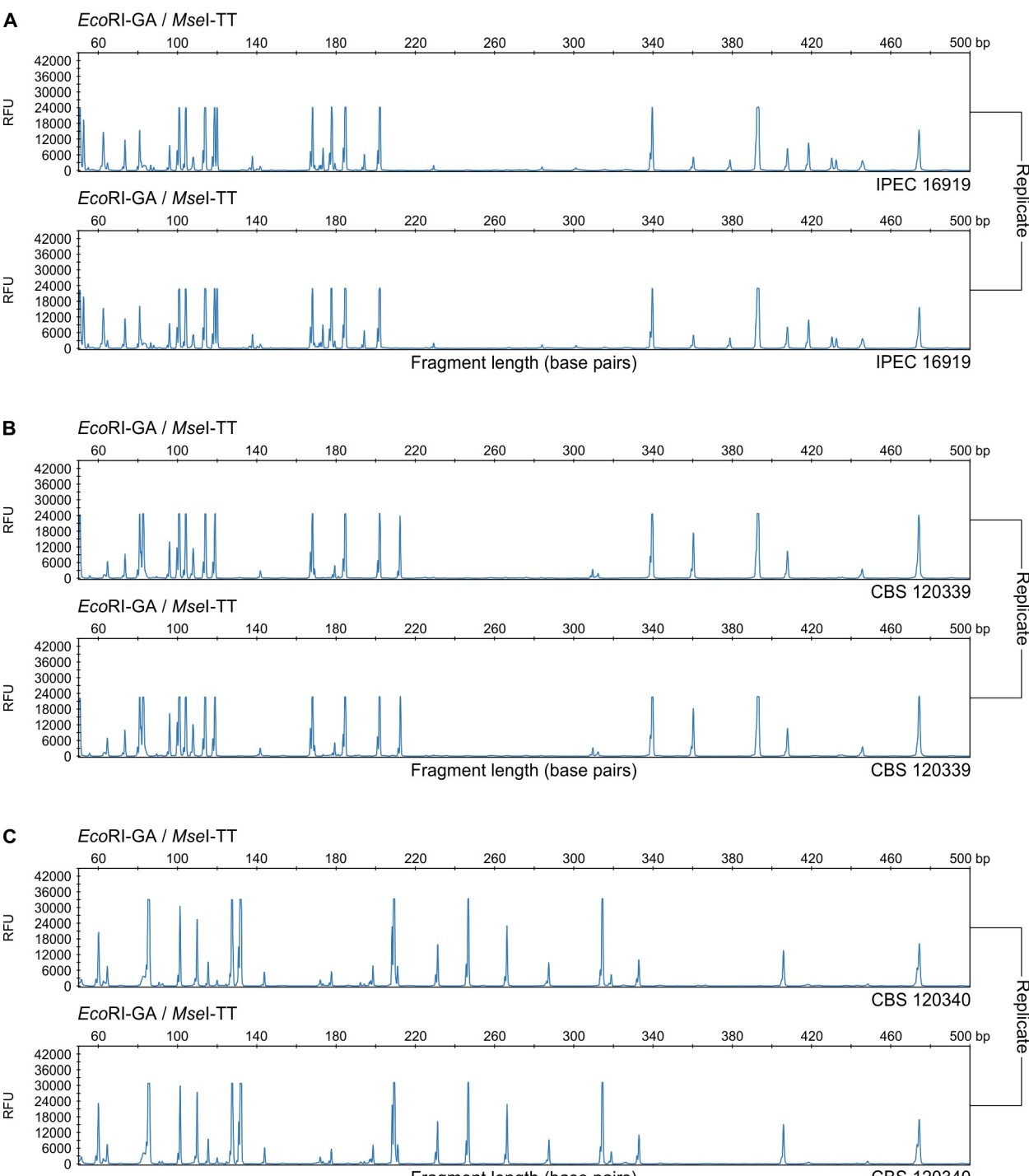

**Fig 4.** Electropherograms from an ABI3100 depicting the range of fragments between 50 and 500 bp for *S. brasiliensis* IPEC 16919 (A), *S. brasiliensis* CBS 120339 (B) and *S. globosa* CBS 120340 (C), with *Eco*RI-GA and *Mse*I-TT selective primers labeled with blue (6-FAM). Error rates were never greater than 1.23%, indicating that our protocol is highly reproducible across *Sporothrix* species.

The AFLP-derived MSTs in Fig 7 essentially confirm the diversity structure of *S. brasiliensis*, *S. schenckii*, and *S. globosa* with the majority of isolates having a unique genotype, in contrast with those found for calmodulin sequencing (Fig 1B). In general, AFLP-derived MSTs

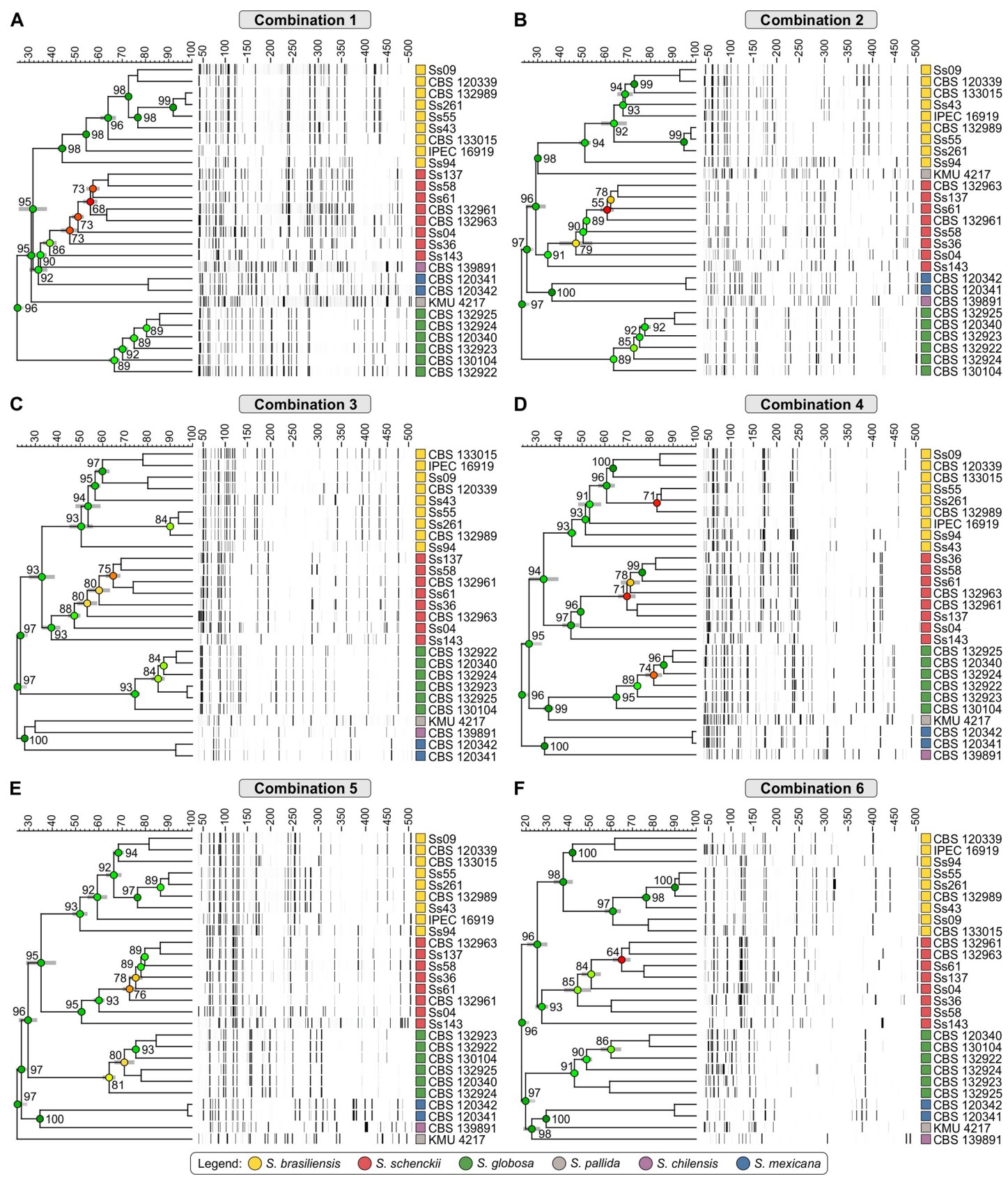

**Fig 5.** Dendrograms of combinations 1 to 6 (A to F, respectively). The dendrograms show the clustering profile of the 27 samples of *Sporothrix*. The dendrograms were constructed by the Jaccard similarity coefficient and UPGMA clustering in the software BioNumerics v.7.6.

were similar in their identification of totally different types of genetic variants. This suggests that despite the enhanced inherent variability of AFLP markers, clusters of isolates remain traceable.

## Discussion

In this study, we develop a new AFLP technique aimed at studying the genetic epidemiology of medically relevant *Sporothrix* species. The most promising result is the discovery of cryptic diversity in species previously thought to be prevalent clonal types, such as *S. brasiliensis*, which is responsible for the cat-transmitted sporotrichosis in Brazil, and *S. globosa*, responsible for large sapronosis taking place in Asia. Our technique, therefore, will enable finer-scale epidemiological patterns to be described than was previously possible.

The first clues that *S. brasiliensis* might have cryptic genetic diversity emerged from host-pathogen interaction data, where genetically identical isolates (based on *CAL* sequences) had distinct virulence profiles when inoculated in BALB/c mice for variables such as animal weight loss, death, and capacity to disseminate to organs [81]. For *S. schenckii* the higher levels of intraspecific genetic variability are associated with more variable virulence profiles in BALB/c [82] suggesting that genotype is an important factor in determining clinically-relevant phenotypes. Considering all molecular markers previously used to explore genetic diversity in *Sporothrix*, including chitin synthase [83], elongation factor 1α [2], β-tubulin [2, 83], ITS1/2+5.8s [20, 78, 79] and calmodulin [2, 47, 83], the last perform best in estimating diversity by showing up to 31.69% variable parsimony-informative sites and comprising for the largest number of genotypes. However, our study shows that calmodulin (and other markers) underestimated the genotypic diversity of *S. brasiliensis*. Therefore, we clearly need new, higher-power, markers to study the recent expansion (2015–2019) of epizooties of feline sporotrichosis due to *S. brasiliensis* towards Northeastern Brazil [1].

Classic studies in the literature describe the use of AFLP markers as one of the most informative and cost-effective DNA fingerprinting methods for genetic characterization of pathogenic *Sporothrix*. The technique usually shows a strong geographical population structure and is useful to speciate *Sporothrix* isolates [20, 39, 40]. Through our markers, we were able to identify *Sporothrix* down to species level, with results similar to the gold standard DNA sequencing of the calmodulin encoding gene [47], *CAL*-RFLP [41], or species-specific PCR [46]. Moreover, we were able to differentiate *Sporothrix* down to strain level, surpassing the resolution provided by the gold-standard method (calmodulin) used to investigate diversity in *Sporothrix*. Our single protocol was easily transferred between distantly related taxa, including members of the clinical and environmental clades.

The main advantages of using AFLP analysis include the use of a standard protocol in combination with different restriction endonucleases and the choice of adding one or more selective nucleotides in the selective EcoRI and MseI primers to achieve optimal fingerprints without prior knowledge of the organism's genome sequence. This is useful for non-model organisms such as *Sporothrix*. Disadvantages include the dominance of alleles, and the possible non-homology of comigrating fragments belonging to different loci, leading to suboptimal reproducibility, particularly across different platforms. However, we took advantage of the growing number of full genome sequences available for *Sporothrix* [58–63] to generate 2,304 virtual fingerprints of *Sporothrix* DNA, reducing time and costs related to extensive trials, minimizing possible reproducibility errors.

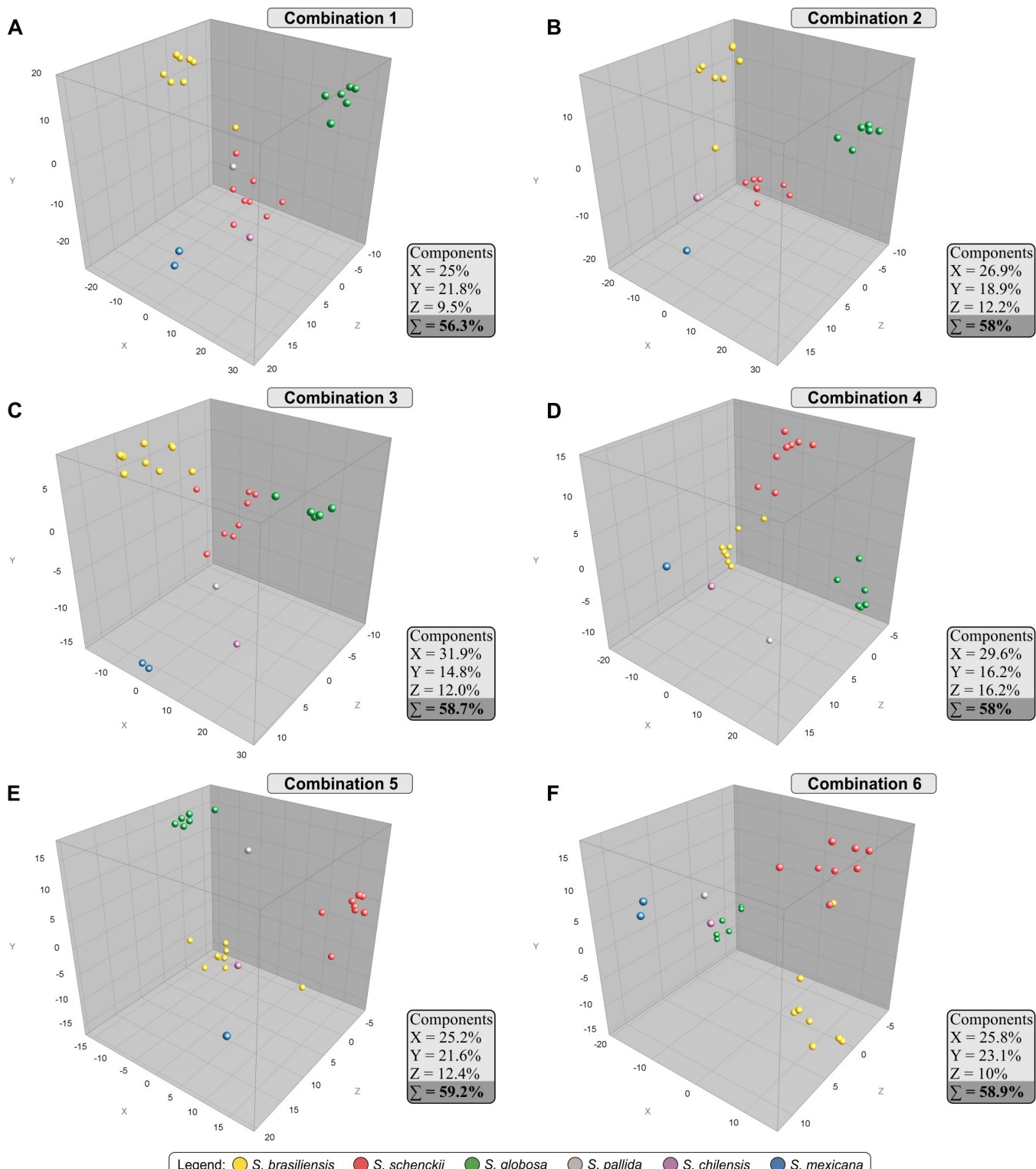

**Fig 6.** PCA of combinations 1 to 6 (A to F, respectively). PCAs were constructed in the software BioNumerics v.7.6. The PCAs were used to demonstrate the correlations among the 27 samples of *Sporothrix* using the six primer pair combinations. The isolates are represented by colored circles.

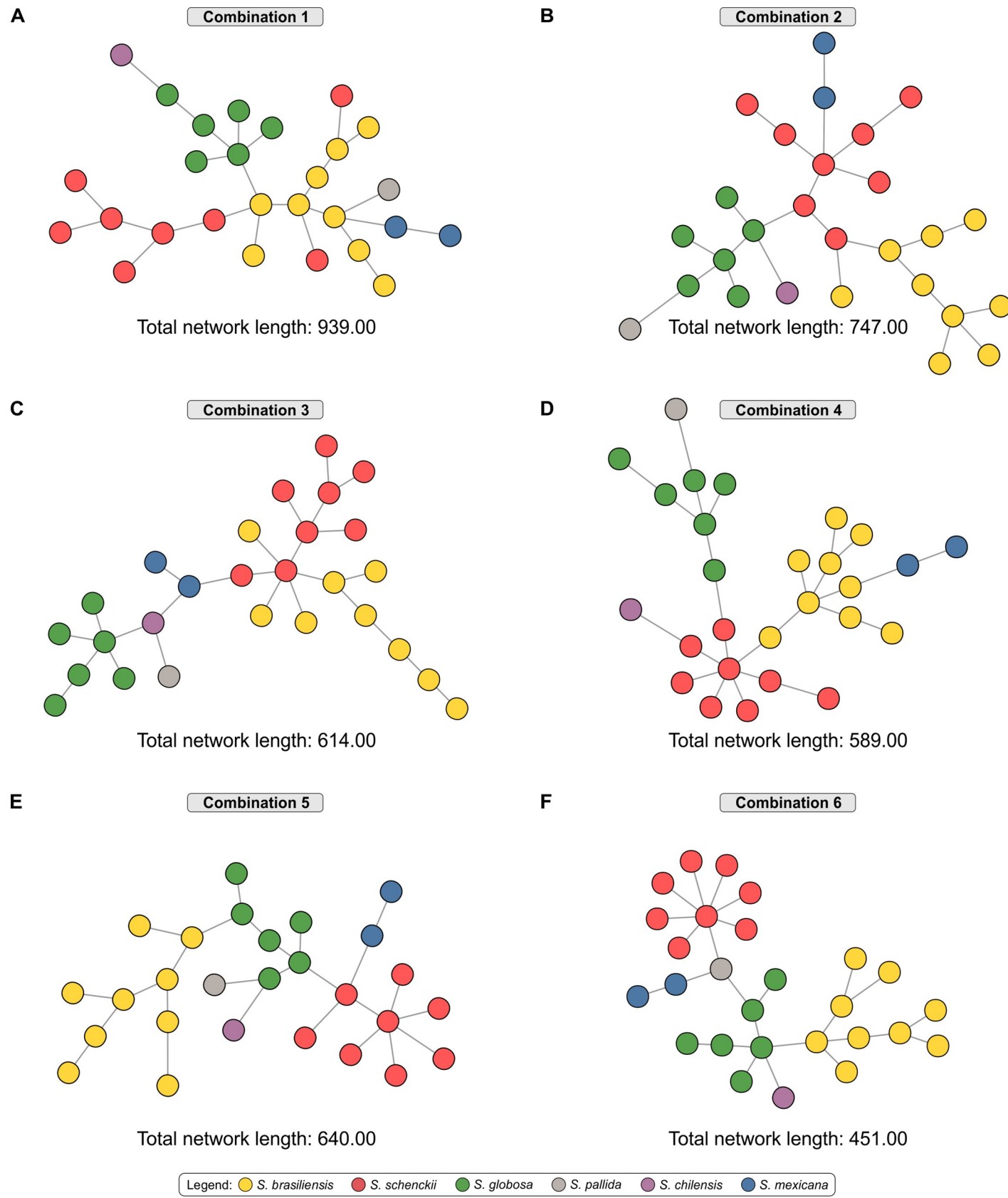

**Fig 7.** AFLP-derived minimum spanning trees (MSTs) of *Sporothrix* isolates using combinations 1 to 6 (A to F, respectively). *CAL* tree and AFLP MSTs were divergent in their identification of different types of genetic variants, since the finer resolution was obtained using AFLP markers.

Therefore, the first step in our standardization included an *in silico* approach using the programs AFLPinSilico [55] and ISIF [56] to optimize the best combination of selective primers (EcoRI+2 and MseI+2) to produce the largest number of polymorphic AFLP markers and address questions related to speciation, genetic diversity, and population structure. The results revealed that, while the two bioinformatic programs had minor divergences between them, *in vitro* results were consistent with *in silico* prediction. The main criterion used was to choose combinations that would reveal cryptic diversity in *S. brasiliensis* and *S. globosa*, two species that were previously characterized as having low genetic diversity or clonal population structure using DNA-sequencing methods [14, 20, 42, 44, 77–79].

The size and complexity of the *Sporothrix* genome can have a tremendous influence on AFLP patterns, due to the amount of DNA that is necessary for the initial restriction/ligation step, as large genomes usually require larger amounts of DNA [84]. Here, 200 ng was enough to produce clear, intense AFLP fragments [85]. Another observation from our *Sporothrix* genome scan follow-up is that the number of selective bases greatly influenced the number and diversity of fragments. Generating too many fragments is not ideal, since it makes it difficult to unambiguously score AFLP electrophoresis profiles (e.g., owing to comigrating non-homologous fragments). If, however, the primer combination generates a low number of AFLP markers, it will reduce the probability of polymorphism detection. We found that two selective bases (EcoRI+2 and MseI+2) produced the ideal number of AFLP markers (i.e., 96–137 polymorphic fragments) to explore diversity in *Sporothrix*. This holds true also when dealing with microorganisms having a larger genome size [29]. We found a higher number of AFLP bands *in silico* in at least 62.5–70.7% of combinations for *S. pallida* (genome size of ~37.8 Mb) compared to *S. brasiliensis*, *S. schenckii* or *S. globosa* (32.5–33.4 Mb) [85] suggesting that, as expected, AFLP patterns scale in complexity with genome size.

Understanding the spread of *Sporothrix* species by exploring inter- and intraspecific genetic diversities is fundamental to tackle the increasing number of cases in recent years [86]. Many molecular techniques have been employed for *Sporothrix* to address these questions [20, 39–41, 46, 85, 87, 88], and have been successful in providing quality data. However, limitations include the low number of polymorphic and reproducible characters evaluated. AFLP has been considered to be highly reproducible and discriminatory, mainly for epidemiological studies [89], and can be employed for any organisms without the need of sequence information [90]. However, only Zhang *et al*. employed the technique on *S. brasiliensis*, but they were not able to describe high levels of diversity [20]. Our *in vitro* AFLP succeeded, demonstrating it is a relevant method for epidemiological studies of *Sporothrix*, which is in line with previous studies that used the technique [20, 39, 40]. Comparing haplotype and AFLP dendrograms or AFLP-derived MSTs generated here, the DNA fingerprint methods demonstrated meaningful variations in *Sporothrix* spp.

DNA-sequencing data demonstrated there are at least two different *S. brasiliensis* populations circulating in Brazil, one belonging to the Rio Grande do Sul cluster and the other related to the long-lasting epidemic taking place in Rio de Janeiro, which is spreading to neighboring states such as São Paulo, Minas Gerais and Espírito Santo [44, 86]. Zhang *et al*. [20] demonstrated that *S. brasiliensis* clustered into three different clades in an AFLP study, and one of the clades contained isolates from Southern Brazil, agreeing with the phylogeography proposed by Rodrigues *et al*. [44]. The same pattern was evidenced in our study, and strains from Rio Grande do Sul (e.g., Ss55, Ss261, and CBS 132989) presented an identical clustering pattern,

showing high levels of similarity among the isolates, which differed from those originated from other Brazilian regions. The three clusters of *S. brasiliensis* found by Zhang *et al.* [20] were supported by high bootstrap values in phylogenetic analysis (i.e., partial *CAL*, *TEF1*, and *TEF3*), but the AFLP fingerprints showed little variation [20]. Our fingerprints revealed meaningful variations in *S. brasiliensis*, with a large number of polymorphic bands, showing that the strategy used here based on extensive *in silico* screening of selective bases was fundamental to guide the choice of the best primer combination.

Among the pathogenic species, *S. schenckii* is considered to have the highest genetic diversity [20, 47, 83], with evidence of genetic recombination [42]. Diversity in *S. schenckii* was also found by other researchers, who separated this species into five groups using AFLP [20]. Our AFLP setup was able to detect diversity within *S. schenckii*.

AFLP markers revealed that *S. globosa* also presented subtle diversity, but it is important to point out that these isolates were less diverse than *S. schenckii* and *S. brasiliensis*, confirming previous findings of Zhang *et al.*, who considered that the DNA fingerprints were identical in *S. globosa*. *Sporothrix globosa* is widely distributed in temperate and warm regions [47, 83, 91], and rapid dispersal by unknown vectors could explain the similar genotypes for isolates collected from distant geographical regions [20, 92]. In addition, Zhao *et al.* [40] also detected cryptic diversity in *S. globosa* evaluating a higher number of strains, similar to the variation found using our new AFLP markers.

Comparison of the DNA-sequence tree with AFLP dendrogram using the congruence index (I*cong*) revealed a strong topological congruence between any two AFLP dendrograms or between AFLP dendrograms and a calmodulin tree, supporting the use of all combinations to distinguish *Sporothrix* species. The congruence observed is in agreement with previous studies, which found that *S. brasiliensis*, *S. schenckii* and *S. globosa* are related in a pathogenic/clinical clade and *S. mexicana*, *S. pallida*, and *S. chilensis* are nested in an environmental clade, supported by high bootstrap values or cophenetic values [2, 42–44, 79]. On one hand, several studies have demonstrated that moderate numbers of AFLP fragments are necessary to recover the correct topology of a DNA-sequence tree with high bootstrap support values (i.e. >70%) [93–95]. On the other hand, a higher number of taxa (i.e., covering all 53 *Sporothrix* species described so far) may increase the number of possible trees and reduce internode distances for a given tree length, making it less likely to recover the correct tree [95, 96]. However, this size effect is more likely to occur in inferences considering ancient radiations [95, 96], which does not seem to be the case of medically relevant *Sporothrix* species, especially *S. brasiliensis*, which likely originated from a recent radiation event (about 3.8–4.9 MYA) based on its geographical distribution and phylogenetic inferences [42, 58, 97].

Using our new AFLP markers, we can now better track routes of disease transmission during epizooties and zoonosis in Brazil, allowing the possibility of linking specific genotypes to antifungal susceptibility profiles as well as addressing links between clinical outcomes in feline and human sporotrichosis. All six primers pairs performed well, but the most precise level of inter-species discrimination and the highest level of intra-species discrimination of the *Sporothrix* isolates were observed in the AFLP EcoRI-FAM-GA/MseI-TT, EcoRI-FAM-GA/MseI-AG and EcoRI-FAM-TA/MseI-AA sets. These combinations stand out among all combinations for having the best diversity indices and the lowest error rates, and thus may be valuable in human and veterinary diagnostics as well as in epidemiology. Moreover, our DNA fingerprinting assay can be further transferred between laboratories to give insights into the ecology and evolution of pathogenic *Sporothrix* species and to inform management and mitigation strategies to tackle the advance of sporotrichosis. Although sporotrichosis has a global distribution, pathogenic species are not evenly distributed and individual lineages that vary in pathogenicity still occur in geographically limited ranges, such as *S. brasiliensis*. Thus, as

*Sporothrix* genotypes continue to expand their range, we need to consider using genome-wide markers to identify, distinguish, and track these emerging pathogens spreading through the mammal hosts.

## Supporting information

**S1 Table. Diversity of AFLP fragments (50–500 bp) based on *in silico* characterization of *Sporothrix* spp. genomes for 256 selective combinations.**
(PDF)

**S2 Table. Cluster similarities among all the combinations evaluated.**
(PDF)

## Author Contributions

**Conceptualization:** Anderson Messias Rodrigues.

**Data curation:** Jamile Ambrósio de Carvalho, Anderson Messias Rodrigues.

**Formal analysis:** Jamile Ambrósio de Carvalho, Anderson Messias Rodrigues.

**Funding acquisition:** Zoilo Pires de Camargo, Anderson Messias Rodrigues.

**Investigation:** Jamile Ambrósio de Carvalho, Anderson Messias Rodrigues.

**Methodology:** Jamile Ambrósio de Carvalho, Anderson Messias Rodrigues.

**Project administration:** Anderson Messias Rodrigues.

**Software:** Jamile Ambrósio de Carvalho, Anderson Messias Rodrigues.

**Supervision:** Zoilo Pires de Camargo, Anderson Messias Rodrigues.

**Visualization:** Jamile Ambrósio de Carvalho.

**Writing – original draft:** Jamile Ambrósio de Carvalho, Anderson Messias Rodrigues.

**Writing – review & editing:** Jamile Ambrósio de Carvalho, Ferry Hagen, Matthew C. Fisher, Zoilo Pires de Camargo, Anderson Messias Rodrigues.

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
