## [Decision Letter · Decision Letter 0]

20 Mar 2020

Dear Dr. Rodrigues,

Thank you very much for submitting your manuscript "Genome-wide mapping using new AFLP markers to explore intraspecific variation among pathogenic Sporothrix species" for consideration at PLOS Neglected Tropical Diseases. As with all papers reviewed by the journal, your manuscript was reviewed by members of the editorial board and by several independent reviewers. The reviewers appreciated the attention to an important topic. Based on the reviews, we are likely to accept this manuscript for publication, providing that you modify the manuscript according to the review recommendations. 

Sincerely,

Todd B. Reynolds

Deputy Editor

Todd Reynolds

Deputy Editor

Reviewer's Responses to Questions

**Key Review Criteria Required for Acceptance?**

**Methods**

-Are the objectives of the study clearly articulated with a clear testable hypothesis stated?

-Is the study design appropriate to address the stated objectives?

-Is the population clearly described and appropriate for the hypothesis being tested?

-Is the sample size sufficient to ensure adequate power to address the hypothesis being tested?

-Were correct statistical analysis used to support conclusions?

-Are there concerns about ethical or regulatory requirements being met?

Reviewer #1: The Methods meet all criteria.

Reviewer #2: An ongoing cat-transmitted Sporotrichosis epidemic in Brazil has stimulated over the past few years research interest in this disease that was largely neglected for the previous 50 years, apart from some reports on individual cases or a few small outbreaks usually linked to contaminated plant material. DNA sequences have revealed that the causal agents of sporotrichosis can be any of several Sporothrix species. There are clear differences in symptoms, mode of transmission, virulence, antibiotic resistance, and geographic distribution of these species. Based on sequences of a limited number of genes, it is also well known that there is intraspecific variation between strains of the same species. With the expansion of the cat-transmitted outbreak in Brazil from Rio de Janeiro to other cities, the risk of the disease being spread to other countries is increasing. In addition, intraspecific genetic variation can be used in population style analytics to indicate origins and routes of movement of fungi. For these reasons accurate and fast diagnosis of the causal agents, down to a specific haplotype, is not only highly informative for researchers, but often crucial in terms of selecting the most appropriate treatment.

The aim of the present study was to develop a DNA barcoding technique that not only could facilitate accurate and fast diagnosis, but also to explore and intraspecies variation. The authors have utilizied the available genome sequences of six Sporothrix species to develop new AFLP markers not only to accurately identify species, but also to assign isolates to intraspecies groups. The study was done with extreme care and thoroughness and I could not find any flaws in experimental design, laboratory techniques, analyses, or the interpretation of data. In addition, the paper was carefully written and thoroughly edited, leaving me as a reviewer with very little to comment on. I wish more manuscripts submitted for review would be so carefully edited.

In the attached pdf I have made some minor suggestions and corrections.

Reviewer #3: The objective, experimental design, samples and data analysis were clearly stated and presented.

**Results**

-Does the analysis presented match the analysis plan?

-Are the results clearly and completely presented?

-Are the figures (Tables, Images) of sufficient quality for clarity?

Reviewer #1: The results are well drafted and clearly illustrated.

Reviewer #2: No problems here, see above.

Reviewer #3: Results were clearly presented in details.

**Conclusions**

-Are the conclusions supported by the data presented?

-Are the limitations of analysis clearly described?

-Do the authors discuss how these data can be helpful to advance our understanding of the topic under study?

-Is public health relevance addressed?

Reviewer #1: -Are the conclusions supported by the data presented? YES 

-Are the limitations of analysis clearly described? YES 

-Do the authors discuss how these data can be helpful to advance our understanding of the topic under study? YES

-Is public health relevance addressed? YES

Reviewer #2: No problems here, see above.

Reviewer #3: Conclusions are adequate.

**Editorial and Data Presentation Modifications?**

Reviewer #1: At line 432 some missing text makes this sentence unclear: "cluster 4, fragments generated varied from 14-44, and S. pallida isolate clustered apart from the S. pallida complex in AFLP combinations #1, #2 and #4." 

The Discussion needs to be trimmed to minimize overlap with the Introduction, e.g., compare lines 94-112 with 556 – 564. The Discussion should not be written as a separate self-contained essay that assumes the reader knows nothing of what's stated in the Introduction and Results.

Reviewer #2: Just a very few minor editorial corrections needed.

Reviewer #3: (No Response)

**Summary and General Comments**

Reviewer #1: Intriguingly novel bioinformatics-based genetic typing technique for epidemiology, and successfully applied.

Reviewer #2: See above

Reviewer #3: This paper reports development of amplified-fragment-length polymorphisms (AFLP) to assess genetic diversity among Sporothrix species. Whole genome sequences from Sporothrix species were used to generate virtual AFLP fingerprints, which guided the development of 6 primer pair combinations to be tested in wet lab. A total of 27 Sporothrix isolates (S. brasiliensis, S. schenckii, S. globosa, S. mexicana, S. chilensis, and S. pallida) obtained from clinical lesions of patients, animals and environmental sources were analyzed by the newly developed AFLP markers. Three AFLP markers revealed high discriminating power in distinguishing different strains of Sporothrix species that were thought belonging to the same clonal lineages. These genetic markers are suitable to track Sporothrix spp transmission during epizooties and zoonosis, and to understand the ecology and evolution of pathogenic Sporothrix species.

The manuscript reads well. The authors took an in-depth analysis of the newly developed AFLP markers and showed their high discriminatory power to differentiate different strains among Sporothrix species.

I only have minor comments:

In Abstract, added the results from the 27 Sporothrix isolates studied. 

In Figure 7, the topologies of Combinations 1 to 4 are similar, but Combinations 5 and 6 are quite different, any explanation?

Whereas it is clear that AFLP has many advantages to be used for epidemiology studies, the author may want to discuss its disadvantages, such as the need of DNA sequencing facility (even though not sequenced in this case), and the uncertainty of amplified fragments with similar size but different sequences. The portability of this method will need to be proved by other labs in future studies.

PLOS authors have the option to publish the peer review history of their article (what does this mean?). If published, this will include your full peer review and any attached files.

Reviewer #1: Yes: Richard Summerbell

Reviewer #2: No

Reviewer #3: No
---

## [Editor Report · Decision Letter 1]

27 Apr 2020

Dear Dr. Rodrigues,

We are pleased to inform you that your manuscript 'Genome-wide mapping using new AFLP markers to explore intraspecific variation among pathogenic Sporothrix species' has been provisionally accepted for publication in PLOS Neglected Tropical Diseases.

Best regards,

Todd B. Reynolds

Deputy Editor

Todd Reynolds

Deputy Editor

---

## [Editor Report · Acceptance letter]

15 Jun 2020

Dear Dr. Rodrigues,

We are delighted to inform you that your manuscript, "Genome-wide mapping using new AFLP markers to explore intraspecific variation among pathogenic Sporothrix species," has been formally accepted for publication in PLOS Neglected Tropical Diseases.

Best regards,

Shaden Kamhawi

co-Editor-in-Chief

Paul Brindley

co-Editor-in-Chief
